# Mapping m^6^A Sites on HIV-1 RNA Using Oligonucleotide LC-MS/MS

**DOI:** 10.3390/mps7010007

**Published:** 2024-01-10

**Authors:** Alice Baek, Asif Rayhan, Ga-Eun Lee, Sarah Golconda, Hannah Yu, Shihyoung Kim, Patrick A. Limbach, Balasubrahmanyam Addepalli, Sanggu Kim

**Affiliations:** 1Center for Retrovirus Research, The Ohio State University, Columbus, OH 43210, USA; baek.71@buckeyemail.osu.edu (A.B.); lee.8950@buckeyemail.osu.edu (G.-E.L.); golconda.2@buckeyemail.osu.edu (S.G.); yu.2127@osu.edu (H.Y.); kim.6754@osu.edu (S.K.); 2Department of Veterinary Biosciences, The Ohio State University, Columbus, OH 43210, USA; 3Infectious Diseases Institute, The Ohio State University, Columbus, OH 43210, USA; 4Translational Data Analytics Institute, The Ohio State University, Columbus, OH 43210, USA; 5Rieveschl Laboratories for Mass Spectrometry, Department of Chemistry, University of Cincinnati, Cincinnati, OH 45221, USA; rayhanaf@mail.uc.edu (A.R.); limbacpa@ucmail.uc.edu (P.A.L.)

**Keywords:** RNA modification, N6-methyladenosine (m^6^A), HIV-1, mass spectrometry, oligonucleotide LC–MS/MS, target RNA enrichment

## Abstract

The biological significance of chemical modifications to the ribonucleic acid (RNA) of human immunodeficiency virus type-1 (HIV-1) has been recognized. However, our understanding of the site-specific and context-dependent roles of these chemical modifications remains limited, primarily due to the absence of nucleotide-resolution mapping of modification sites. In this study, we present a method for achieving nucleotide-resolution mapping of chemical modification sites on HIV-1 RNA using liquid chromatography and tandem mass spectrometry (LC–MS/MS). LC–MS/MS, a powerful tool capable of directly analyzing native RNAs, has proven effective for mapping RNA modifications in small RNA molecules, including ribosomal RNA and transfer RNA. However, longer RNAs have posed challenges, such as the 9 Kb HIV-1 virion RNA, due to the complexity of and ambiguity in mass differences among RNase T1-cleaved RNA fragments in LC-MS/MS data. Here, we introduce a new target RNA enrichment method to isolate small local RNA fragments of HIV-1 RNA that potentially harbor site-specific N6-methyladenosine (m^6^A) modifications. In our initial trial, we used target-specific DNA probes only and encountered insufficient RNA fragmentation due to inefficient S1 digestion near the target site. Recognizing that inefficient S1 digestion by HIV-1 RNA is likely due to the formation of secondary structures in proximity to the target site, we designed multiple DNA probes annealing to various sites of HIV-1 RNA to better control the structures of RNA substrates for S1 digestion. The use of these non-target DNA probes significantly improved the isolation of more homogeneous target RNA fragments of approximately 50 bases in length. Oligonucleotide LC-MS/MS analysis of these isolated target RNA fragments successfully separated and detected both m^6^A-methylated and non-methylated oligomers at the two m^6^A-predicted sites. The principle of this new target enrichment strategy holds promise and should be broadly applicable to the analysis of any lengthy RNA that was previously deemed infeasible for investigation using oligonucleotide LC-MS/MS.

## 1. Introduction

Chemical modifications on RNA, collectively known as epitranscriptomes, have been demonstrated to influence various aspects of RNA, including stability, folding, translocation, alternative splicing, and translation [1]. The dysregulation of epitranscriptomes has been implicated in more than 100 human diseases [2,3,4]. Since their initial discovery in the late 1950s, over 300 different types of chemical modifications to RNA have been identified [3,5]. However, we have only recently begun to comprehend the global distribution of these modifications thanks to advances in sequencing technologies [6].

Most studies to date have utilized short-read sequencing methods, involving the reverse transcription of fragmented RNA and PCR amplification of reverse-transcribed cDNA, providing only low-resolution and population-average values of chemical modifications. While useful in identifying chemical modification sites without prior knowledge, short-read data are complicated by potential biases introduced during antibody-mediated target RNA enrichment and cDNA preparation [6,7]. Due to the lack of nucleotide-resolution mapping of site-specific modifications, our understanding of chemical modifications is mostly based on cellular perturbation (overexpression or knockout/knockdown) of host effectors [6,8,9,10,11,12,13,14,15], overlooking the site-specific roles of the epitranscriptome [2,16,17,18,19,20,21]. These studies often reported disparate effects of epitranscriptome depending on gene type, cellular status and viral species [1,9,22].

Mass spectrometry (MS) is a powerful tool, capable of directly analyzing chemical modifications of native RNA molecules by measuring the mass shift of RNA oligomers [23,24]. To identify an unknown modification (a new type of chemical modification), one can employ complete hydrolysis of RNA before LC-MS [25]; however, this approach does not provide modification-site information. Oligonucleotide LC-MS/MS, however, identifies both the positions and types of chemical modifications at single-nucleotide resolution. In this technology, chemical modification sites can be identified via nucleobase-specific digestion of RNA molecules via guanosine (G)-specific ribonuclease T1, cytidine (C)-specific Cusativin, or uridine (U)-specific MC1 ribonucleases, followed by liquid-chromatography-mediated separation of digested products and tandem mass spectrometry of separated oligonucleotides [23,24,26]. Selective cleavage of RNA with one of these nucleobase-specific RNases generates RNA fragments simple enough for MS to detect. These RNA fragments possess unique sequences and display mass differences between chemically modified and unmodified status. This technology can determine chemical modifications in RNAs with known sequences. However, the positions and types of modifications can be identified without prior information [23,24,26]. This direct RNA analysis is also free from the potential technical biases associated with short-read sequencing methods [6,7].

HIV-1 RNAs have significantly higher epitranscriptomic modifications than cellular RNA [12,27]. However, the precise roles of these modifications in HIV-1 replication and their evolutionary benefits remain unclear, exhibiting both pro- and anti-viral effects depending on the experimental conditions and tested replication stages [9,15,28,29,30,31]. Despite frequent cleavage with nucleobase-specific RNases, the analysis of small RNAs, such as transfer RNA (tRNA) and ribosomal RNA (rRNA) with oligonucleotide LC–MS/MS, has proven effective [7,24,32,33,34,35,36,37,38]. However, the analysis of larger RNAs, such as the 9 Kb HIV-1 RNA, remains nearly impossible due to the ambiguity in mass differences among RNase-cleaved fragments in LC-MS/MS data. For example, RNase T1 digestion generates many RNA fragments with the same molecular weight, originating from different parts of the HIV-1 RNA. This prevents the unambiguous mapping of RNase T1-cleaved fragments. In this study, we developed a new method that can effectively enrich target RNA fragments within a mixed pool of long RNA strands. With the new method, we fragmented 9 Kb HIV-1 virion RNA via S1 nuclease digestion and selectively enriched and purified small (~50 bases) target RNA fragments with sufficient purity for oligonucleotide LC-MS/MS analysis. The MS data confirmed the precise locations of two dominant m^6^A modifications on HIV-1 RNA.

## 2. Materials and Methods

### 2.1. HIV-1 Viral RNA Extraction

HIV-1 virion RNAs were extracted from HEK293T cells transfected with the HIV-1 proviral DNA construct, pNL4-3, following the procedures described previously [39] (Figure 1a). Briefly, we transfected HEK293T cells with pNL4-3 using polyethylenimine (PEI) following the manufacturer’s instructions. The cell culture medium was exchanged with fresh medium at 6 h post-transfection and the supernatant was harvested at 72 h for virion RNA extraction. To purify viral particles, the supernatant was subjected to a 10% sucrose gradient centrifugation at 28,000× *g* at 4 °C for 1 h 40 min. After discarding the supernatant, the pellet was resolved in 160 μL 1X HBSS and treated with DNase I treatment (12 U/reaction; New England Biolab, Ipswich, MA, USA) for 30 min at 37 °C. Virion RNA was extracted using TRIzol^TM^ (Thermo Fisher, Waltham, MA, USA) following the company-provided standard protocol. The RNA pellet after the 70% ethanol precipitation was air-dried in the hood and then resuspended in 30 μL of DEPC-treated water followed by the addition of RNase inhibitor (New England Biolabs; final concentration 1 U/uL).

### 2.2. HIV-1-Specific Oligos

We used two sets of DNA oligonucleotide to enrich the target RNA fragments: (i) biotinylated, target-specific oligomers (b-target: /5′-GCU ACA AGG GAC UUU CCG CUG GGG ACU UUC CAG GGA G-3′; Integrated DNA Technologies) and (ii) non-target oligomers specific to 110 different sites of HIV-1 at least 40–50 bases away from each other (non-target oligo sequences are available in Appendix A).

### 2.3. S1 Nuclease Digestion and the Recovery of Biotinylated-DNA/RNA Duplex

To enrich target RNA, 10 µg of HIV-1 virion RNA was initially incubated with either b-target oligos only (HIV-1 RNA:DNA oligo molar ratio of 1:100) or a mixture of b-target oligos and non-target oligos (at an HIV-1 RNA:DNA molar ratio of 1:30) at 65 °C for 5 min in 0.5X nuclease-free duplex buffer (Integrated DNA Technologies; 30 mM HEPES, pH 7.5; 100 mM potassium acetate) in heat block and then slowly cooled to room temperature. It was followed by nuclease S1 digestion (Invitrogen; final concentration 2.5 U/uL). The sample was briefly mixed by tapping and left for 2 h at room temperature for RNA digestion. The S1-digested RNA samples were purified using phenol–chloroform [41] and eluted in 40 uL TE buffer.

The target DNA:RNA duplexes with b-target oligos were recovered using Streptavidin beads (Dynabeads™ MyOne™ C1, Thermo Fisher Scientific) following the manufacturer’s instructions. Briefly, the S1-digested RNA sample was mixed with the beads, resuspended in 2X binding and washing buffer (equal volume with the sample,) and incubated for 10 min at room temperature on HulaMixer™ Sample Mixer (Thermo Fisher) in vibration mode (100 rpm). The streptavidin-RNA sample was then put onto a magnetic stand until the solution became clear and supernatant was discarded. The sample was washed twice with 1X binding and washing buffer by repeating this step. The beads were then resuspended with 20 µL of 4 µM biotin solution (in 0.5% dimethylsulfoxide), briefly mixed via tapping, and left at room temperature for 10 min. The sample tube was briefly spun down and put onto a magnetic stand until it became clear (~5 min). The biotinylated-DNA/RNA (b-DNA/RNA) duplex was then recovered by carefully transferring the supernatant to a new tube. The presence and the size of b-DNA/RNA duplex was analyzed on 12% polyacrylamide gel electrophoresis.

### 2.4. Oligonucleotide LC-MS/MS of HIV-1 RNA Fragment

For RNase T1 digestion, about 200 ng of the purified target b-DNA/RNA duplex (see 2.3 above) was initially denatured at 95 °C for 2 min, snap cooled at 4 °C, and digested with 50 units of RNase T1 (Worthington, Columbus, OH, USA) at 37 °C for 2 h. The digested sample was then dried in a speedvac system (Thermo Scientific). LC-MS/MS analysis was performed using a BEH C18 column (1.7 µm, 0.3 × 150 mm, Waters, Milford, MA, USA) with Ultimate 3000 Ultra-High-Performance Liquid Chromatography (UHPLC; Thermo Scientific) coupled to a Synapt G2-S (Quadrupole Time-of-Flight, Q-TOF, Waters, Milford, MA, USA) mass spectrometer. Gradient chromatography was conducted with a flow rate of 5 µL min^−1^ using mobile phase A (8 mM TEA and 200 mM HFIP, pH = 7.8 in water) and mobile phase B (8 mM TEA and 200 mM HFIP in 50% methanol) at 60 °C. The gradient included an initial hold at 3% mobile phase B for sample loading, followed by a ramp to 55% mobile phase B over 70 min, reaching 99% in 2 min with a 5 min hold before re-equilibration (30 min) at 3% mobile phase B for initial conditions. The digestion products in the chromatographic eluent were detected in negative ion mode via electrospray ionization (ESI) on a Synapt G2-S (Quadrupole Time-of-Flight, Q-TOF). The mass spectrometer was operated in V-mode (sensitivity mode). The ESI conditions involved 2.2 kV at capillary, 30 V at sample cone, with source and desolvation temperatures maintained at 120 °C and 400 °C, respectively, and gas flow rates at 3 and 600 L h^−1^, respectively. A scan range of 545–2000 *m*/*z* (0.5 s) was employed for first-stage (MS) data acquisition and 250–2000 *m*/*z* (1 s) for second (MS/MS)-stage data acquisition. The first stage of MS identifies the mass of the oligonucleotide, while the second stage helps to identify the nucleotide sequence based on sequence-informative fragment ion patterns. The top three most abundant ions in the first stage were chosen to face fragmentation for MS/MS using an *m*/*z*-dependent collision energy profile (20–23 V at *m*/*z* 545; 51–57 V at *m*/*z* 2000) before exclusion for 60 s using the dynamic exclusion feature.

### 2.5. Synthetic RNA Control

As a positive control experiment for oligonucleotide LC-MS/MS, we used two synthetic RNA oligos each harboring m^6^A at A8975 (m^6^A8975: GAA CUG CUG ACA UCG AGC UUG CUA CAA GGG **m^6^A**CU UUC CGC UGG GGA CUU UCC AGG GAG GCG UGG CCU GGG CGG GAC UGG GGA; Dharmacon) and at A8989 (m^6^A8989: GAA CUG CUG ACA UCG AGC UUG CUA CAA GGG ACU UUC CGC UGG GG**m^6^A** CUU UCC AGG GAG GCG UGG CCU GGG CGG GAC UGG GGA; Dharmacon). These synthetic oligos were used in parallel with the HIV-1 RNA fragments in “Oligonucleotide LC-MS/MS of HIV-1 RNA fragment” above.

### 2.6. LC-MS/MS Data Processing

Mongo Oligo mass calculator (http://mass.rega.kuleuven.be/mass/mongo.htm; accessed on 11 November 2021) was used to predict the *m*/*z* values for RNase T1 digestion products (and their fragment ions) of a 40-base-long HIV-1 RNA sequence. To manually identify and assign the m^6^A modification, a ~14 Da mass shift was scored for the theoretically expected oligonucleotides, considering cleavage at the 3′-end of guanosine, in the modified RNA compared to the unmodified counterpart. Control experiments were conducted using synthetic oligos, incorporating m^6^As at either A8975 or A8989 to attribute the m^6^A modification to positions 8975 and 8989, respectively.

## 3. Results

### 3.1. Identification of Potential Site-Specific m^6^A Modifications on HIV-1 Viral RNA

Several common chemical modifications, including m^6^A, 5-methylcytosine (m^5^C), 2′-O-methylation (Nm), and N4-acetylcytidine (ac^4^C), have been mapped onto the HIV-1 genome using short-read sequencing methods [10,12,13,14,30,42]. These studies, however, provide only low-resolution, fragmented information, typically spanning approximately 40 to 200 bases, about modification sites. Due to the lack of nucleotide-resolution data, the site-specific and context-dependent roles of RNA modifications remain largely unknown for HIV-1.

We recently developed novel Nanopore Direct RNA Sequencing (DRS) methods for full-length HIV-1 RNA analysis and demonstrated potential chemical modification sites on full-length HIV-1 RNAs [40]. Our DRS data revealed three predominant site-specific modification signals that coincide with m^6^A sequence motifs (DRACH: D = A/G/U, R = A/G, H = A/C/U) at positions A8079, A8975, and A8989 within the 3′ untranslated region (U3) of HIV-1 RNA [40]. Using m^6^A-detection tools, such as m6Anet [43] and Nanom6A [44], these three sites were identified as potential m^6^A modification sites (Figure 1b) [40].

### 3.2. Enriching Target RNA Fragments of HIV-1 RNA for Oligonucleotide LC-MS/MS

To validate the potential site-specific m^6^As identified with Nanopore DRS, we employed oligonucleotide LC-MS/MS. Given the challenges for the analysis of full-length.

Examining HIV-1 RNA, we developed a new method to enrich target RNA fragments for oligonucleotide LC-MS/MS. The local RNA containing the target sites needs to be fragmented and enriched to a nearly homogeneous state, sufficient for RNase T1 treatment. The aim is to yield unique-sized RNA isomers that oligonucleotide LC-MS/MS can distinguish. To achieve this, we designed a 37-base biotinylated DNA probe (b-target oligo) to selectively purify small target RNA fragments that harbor potential m^6^A sites (A8975 and A8989) (Figure 1c). An S1 digestion of b-target-probe-bound HIV-1 RNA, followed by streptavidin-bead isolation, enriched target RNA/DNA duplexes (b-DNA/RNA) suitable for oligonucleotide LC-MS/MS. Initial trials for target enrichment using the b-target-oligo probe alone substantially reduced RNA fragments in general (Figure 2(ai)), but the majority of enriched RNAs were substantially larger than expected, heterogeneous in length distribution, and ranged from 75 to 300 bases, indicating incomplete S1 digestion of RNA near the target site (Figure 2b).

### 3.3. Enhanced Enrichment of More Homogeneous Target RNA Fragments Using Non-Target Probes

Given the complex secondary and tertiary structures of HIV-1 RNA structures [45,46], we reasoned that the enrichment of the unexpectedly lengthy and heterogeneous RNAs may result from poor S1 digestion near the target sites due to local secondary structures (Figure 2(ai)). S1 digestion is inefficient for double-stranded RNA [47]. To test this, we designed a set of DNA probes (for a total of 110 non-target probes, see Appendix A) that anneal to different sites of HIV-1 RNA in order to better control the structure of RNA substrates for S1 digestion. These non-target probes were designed to interfere with the formation of local secondary structures based on data from a previous SHAPE assay [46] and leave at least 40–50 bases of single-stranded areas between the probe binding sites (Figure 2a,b). The two non-target probes (NL9360R and NL9530R, see Appendix A) nearest to the target site align approximately 57–58 bases away from the b-target oligo binding sites. These single-stranded regions are favorable targets for S1 digestion [47]. Indeed, when we used a mixture of both b-target oligos and a set of total 110 non-target probes (Appendix A) to release the local structure of HIV-1 RNA, the S1-digestion and streptavidin-bead purification process resulted in more homogeneous b-DNA/RNA duplex bands near 50-base marker DNA on gel electrophoresis, indicating a substantial improvement in our target enrichment process (Figure 2(bii)). The streptavidin-bead-purified b-DNA/RNA duplexes were then subjected to oligonucleotide LC-MS/MS.

### 3.4. Oligonucleotide LC-MS/MS of Synthetic RNA Oligos with m^6^As at A8975 or A8989

As a positive control experiment, we used two synthetic RNA oligos (Figure 3a; see the methods section for the sequences). The sequences of the two synthetic RNA oligos match to the 81 bases that cover positions 8945 to 9025 of HIV-1 RNA. These oligos have fully m^6^A-methylated nucleotides at either 8975A or 8989A. RNase T1 digestion of these oligos, followed by ultra-high-performance liquid chromatography (UHPLC) and mass-spectrometry (MS), revealed the presence of various RNase T1-cleaved oligos of expected sizes, including m^6^A-methylated and non-methylated forms of the two target sites (A8975:ACUUUCCGp, m^6^A8975: m^6^ACUUUCCGp, A8989: ACUUUCCAGp, and m^6^A8989: m^6^ACUUUCCAGp) (Figure 3a). Two cyclic phosphate forms were also detected for the A8975 site (ACUUUCCG>p and m^6^ACUUUCCG>p). As expected, only the m^6^A-methylated A8975 form (m^6^ACUUUCCGp as well as m^6^ACUUUCCG>p) and non-methylated A8989 (ACUUUCCAGp) oligo form were detected for the m^6^A8975 synthetic RNA oligos; on the other hand, for the m^6^A8989 synthetic RNA oligos, the corresponding oligo form for m^6^ACUUUCCAGp and non-methylated versions for A8975 (ACUUUCCGp and/or ACUUUCCG>p) were detected (Figure 3a).

### 3.5. Oligonucleotide LC-MS/MS Confirms the Presence of m^6^As at A8975 and A8989 of HIV-1 RNA

For virion RNA analyses, we isolated approximately 200 ng of target HIV-1 RNA fragments, harboring both A8975 and A8989 sites, following the S1-digestion and streptavidin-enrichment. These RNA were subjected to subsequent RNase T1 digestion and oligonucleotide analysis via LC-MS/MS (Figure 1 and Figure 2). The analysis identified all the expected methylated and non-methylated versions at the two m^6^A-sites (Figure 3b and Table 1). The detection accuracy was ascertained by the retention time and the mass spectral (*m*/*z* values) matches between the HIV-1 RNA samples and synthetic RNA segments. These data therefore validate the presence of m^6^A at positions A8975 and A8989 in HIV-1 RNA. These m^6^As were further confirmed by the reduction in modification signals following m^6^A-eraser (ALKBH5) treatment and through site-directed mutagenesis in an independent study [40].

While LC-MS/MS had the potential to detect any RNA modifications if they were deposited in a site-specific fashion above the detection levels, we did not detect any other modifications in our experiments.

### 3.6. Detection of Both Methylated and Unmethylated Versions of the Same Oligomer Indicates Partial Methylation

Current analysis of the LC-MS/MS data indicates differential modification stoichiometry based on a comparison of the relative abundance of unmodified and modified oligonucleotides (Table 1). The computation of the peak area ratios following baseline noise subtraction estimates that approximately 12% of HIV-1 RNA is methylated at 8975A, and ~63% is methylated at 8989A. More systematic studies to optimize the quantification via LC-MS/MS-based analyses and additional orthogonal assays are needed, however, if we are to accurately determine the stoichiometry of methylation.

## 4. Discussion

Developing methods that enable the analysis of the “true” sequences of native RNA molecules has been a major interest [8]. Liquid chromatography coupled with mass spectrometry (LC-MS) is a powerful tool for modification analysis due to its ability to directly measure the mass values of modified and unmodified RNA molecules. However, its application to long RNAs and weakly abundant transcripts has been challenging. Enriching small, highly homogeneous RNA fragments of target sites can be a way forward but has been known to be a significant challenge for HIV-1 RNA. In this study, we present a novel method that significantly improves target RNA enrichment by effectively fragmenting lengthy HIV-1 RNA using the S1 enzyme and multiple non-target probes. The principles of our method have the potential for broad application, enabling the analysis of lengthy, kilobase-scale RNAs that were previously considered impractical for investigation by oligonucleotide LC-MS analysis.

Our target enrichment strategy notably improved the enrichment and purification of small target fragments of HIV-1 RNA containing both A8975 and A8989 sites. However, this process requires a substantial amount (~10 µg) of 9 Kb full-length virion RNAs to obtain an adequate amount of target RNA for LC-MS analysis. The current sensitivity achieved with this RNA (200 ng of target RNA) was adequate for detecting the presence of m^6^As at these two sites. Nevertheless, accurate quantification of m^6^A modifications would necessitate a significantly larger quantity of viral RNA and a well-designed quantification experiment. Method precision can be further improved by using another ribonuclease enzyme with complementary specificity so that overlapping digestion product is generated to confirm the modification location.

One limitation of this approach is its inability to distinguish nitrogenous base methylation from ribose methylation through oligonucleotide sequencing. This limitation may be addressed by incorporating nucleoside analysis, where base methylation and ribose methylation exhibit different retention times during chromatography. The point of methylation can then be understood through MS/MS analysis of nucleosides.

While this method is effective in analyzing one target section, it has limitations as it retains only the specified target, discarding the remaining sections during the process. Looking ahead, developing a multiplex technique capable of isolating multiple diverse targets from the same RNA sample would be valuable. Such an advancement could broaden the applicability of the approach, enabling the mapping of other regions of HIV-1 RNA or small segments within any long RNA molecules with known sequences.

## Figures and Tables

**Figure 1 mps-07-00007-f001:**
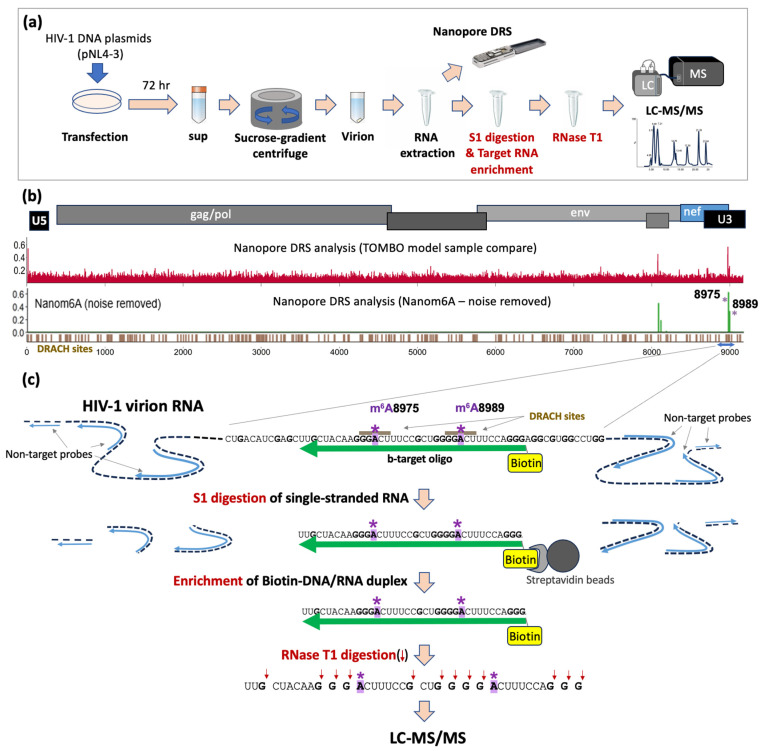
Schematic view of HIV-1 virion RNA analysis. (**a**) HIV-1 virion RNA samples were prepared by transfection of HEK293T cells with HIV-1 proviral DNA plasmids (pNL4-3), followed by sucrose-gradient of supernatant at 72 h post transfection and RNA extraction from purified virions [39]. Virion RNA was then analyzed both by Nanopore Direct RNA Sequencing [40] and RNase T1-mediated oligonucleotide LC-MS for the detection of site-specific chemical modifications. (**b**) Nanopore Direct RNA Sequencing (DRS) of full-length HIV-1 RNA analysis via Tombo–model–sample–compare (Tombo-MSC), and Nanom6A are shown. Both revealed distinct site-specific modification signals at A8975 and A8989 (both denoted by *), coinciding with m^6^A DRACH motifs. (**c**) A schematic view of oligonucleotide LC-MS for the confirmation of m^6^As at A8975 and A8989 sites. The nucleotide sequence of the local area near A8975 and A8989, denoted by the blue arrow in (**b**), is shown. Target RNA fragments, harboring both A8975 and A8989, are enriched via selective S1 digestion on single-stranded RNA sites, followed by streptavidin bead isolation of biotinylated DNA/RNA (b-DNA/RNA) duplex. Blue lines indicate non-target HIV-1 RNA-specific probes (length 20–25 bp) and green lines indicate biotinylated DNA probe (b-target-oligo) binding to the target RNA site. S1 nuclease digestion removes single-stranded RNA, which results in DNA and RNA duplexes. The Biotin-DNA/RNA duplex recovered using streptavidin magnetic beads was subjected to RNase T1 digestion and LC-MS analysis.

**Figure 2 mps-07-00007-f002:**
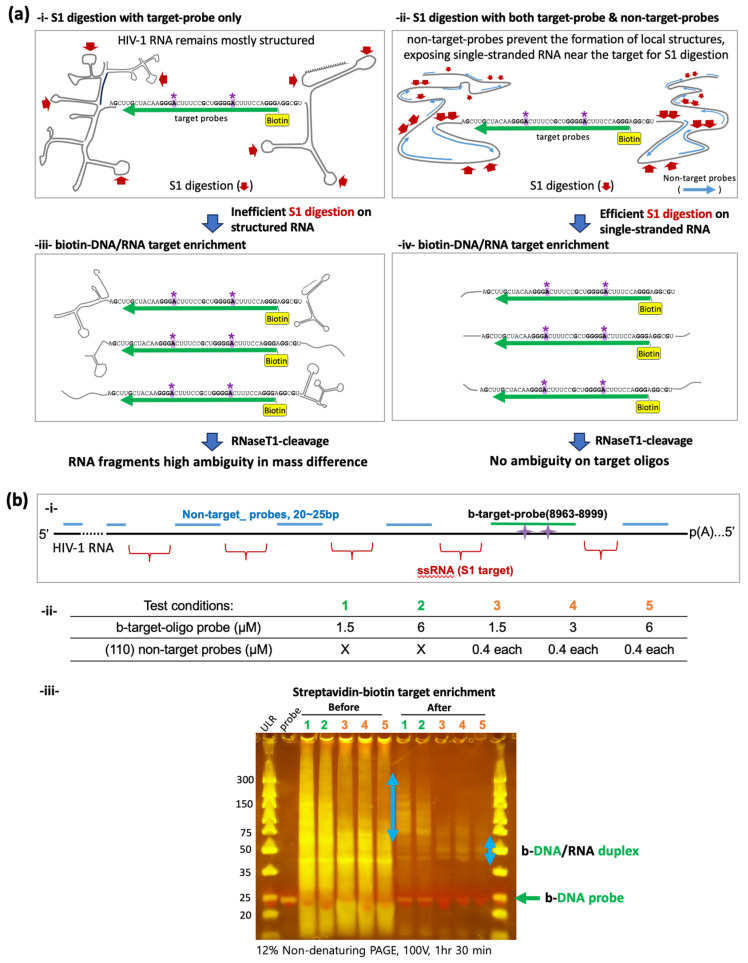
Comparison of S1 digestion with or without non-target probes. (**a**) A schematic view comparing S1 digestion on HIV-1 RNA. Complex RNA secondary structure formation may result in inefficient S1 digestion, leaving unanticipatedly long RNA strands in the enriched biotin-DNA/RNA pool (i,iii). We used 110 different non-target probes to prevent secondary structure formation, which may enhance S1 digestion (ii) and enrich more homogeneous target RNAs (iv). A8975 and A8989 sites are denoted by (*). (**b**) (i) a schematic view of S1-digestion on single-stranded RNA (ssRNA). (ii) Different concentrations of target (b-target-oligo) probes and non-target probes were tested. Conditions 1 and 2 were tested without non-target probes, while 3, 4 and 5 tested with 110 different non-target probes (0.4 µM each). (iii) Non-denaturing PAGE gel electrophoresis before and after the streptavidin–biotin target enrichment. Light blue arrow indicates isolated b-DNA/RNA duplexes. The green arrow indicates unbound DNA probes. In all conditions, streptavidin–biotin enrichment step significantly reduced non-target RNA fragments (before vs. after). After the enrichment, conditions 3, 4, and 5 using non-target probes showed b-DNA/RNA duplex near the 50-base marker, whereas conditions 1 and 2 showed much larger fragments ranging from 75 to 300 bases.

**Figure 3 mps-07-00007-f003:**
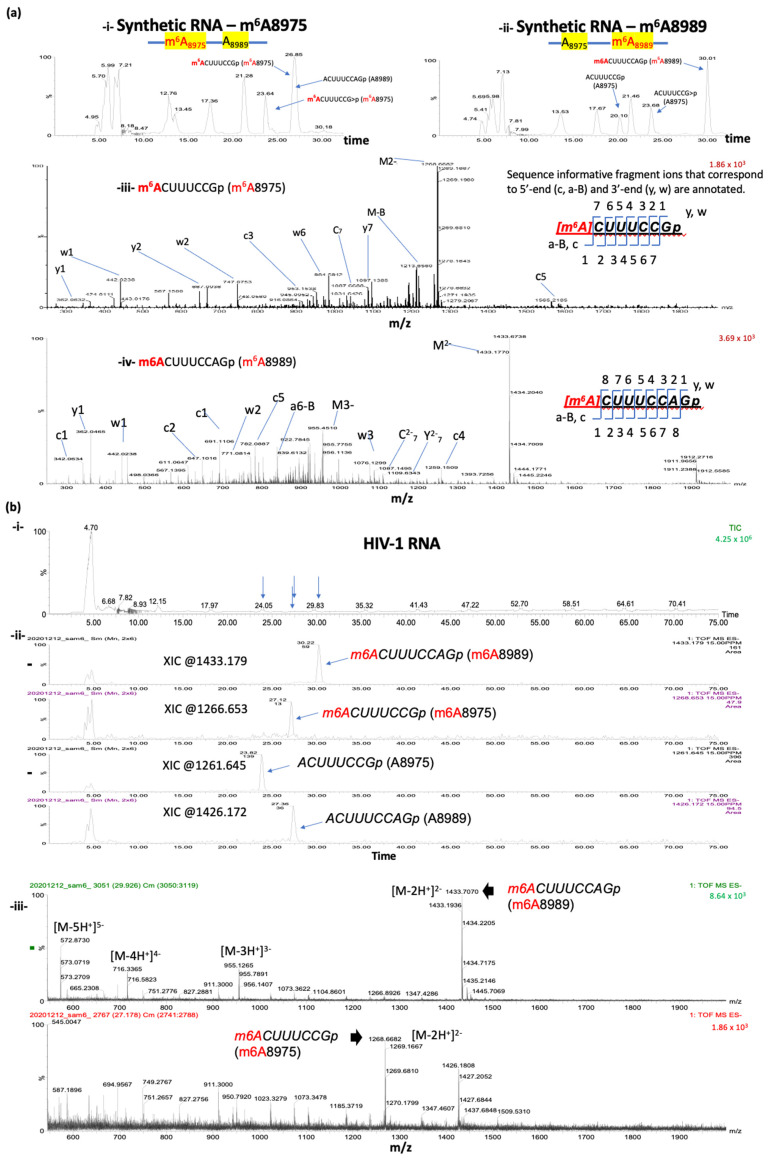
LC-MS/MS based comparison of the RNase T1 digests of target HIV-1 RNA segment with the methylated synthetic oligos of same sequence. (**a**) Detection of RNase T1 digestion products of synthetic oligonucleotides bearing single methylation at either 8975 (i) or 8989 (ii) position in the total ion chromatogram. Mass spectrum of RNase T1 digestion product, m^6^ACUUUCCGp (*m*/*z* 1268.653, mass error < 15 PPM) at position 8975 from synthetic RNA (i,iii) and m^6^ACUUUCCAGp (*1.81* 1433.179, mass error < 15 PPM) at position 8989 (ii,iv). Sequence informative fragment ions that correspond to 5′-end (a-B_n_, c_n_) and 3′-end (y_n_, w_n_) for each of the digestion product are annotated. (**b**) LC-MS analysis of the oligonucleotide segment purified from HIV-1 RNA (i) Total ion chromatogram of the RNase T1 digest. Retention time positions of methylated and unmethylated oligonucleotides are indicated by arrows. (ii) Extracted ion chromatograms for methylated and unmethylated versions of oligomers corresponding to positions at 8975 and 8989. (iii) Mass spectra of the methylated oligonucleotides depicting the multiply charged ions. Note the differences in relative abundance of methylated oligo m^6^ACUUUCCAGp (position 8989, top panel) vs. m^6^ACUUUCCGp (position 8975, bottom panel). Oligonucleotide LC-MS/MS confirmed m^6^A at 8989A and 8975A in HIV-1 viral RNA (iii).

**Table 1 mps-07-00007-t001:** The m^6^A modification frequencies at 8975 and 8989 sites of HIV-1 RNA.

m^6^A Position	Oligomer	Retention Time	Peak Area Counts	Total Area	Relative Ratio of Modified vs. Total
8975	ACUUUCCGp	23.82	434	500	69/569 = 0.1213
	ACUUUCCG>p	20.12	66		
	**m^6^A**CUUUCCGp	27.12	49	69	
	**m^6^A**CUUUCCG>p	23.91	20		
8989	ACUUUCCAGp	27.36	102	102	174/276 = 0.6304
	ACUUUCCAG>p	0	0		
	**m^6^A**CUUUCCAGp	30.22	174	174	
	**m^6^A**CUUUCCAG>p	0	0		

## Data Availability

The data supporting the current study are available from the corresponding authors upon request.

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
