# Peer review of "Mapping m6A Sites on HIV-1 RNA Using Oligonucleotide LC-MS/MS"

_mps, 2024, doi:10.3390/mps7010007_

Round 1

Reviewer 1 Report

Comments and Suggestions for Authors

In this manuscript, the authors developed a new method for targeted oligonucleotide modification identification to confirm the detection result of their previous DRS result on HIV-1 RNA. This is a brand new method for oligonucleotide modification identification, but the drawbacks are also obvious, this method is only suitable for known RNA sequence with known modification on known position. The followings are my comments:

1.       Except for the known RNA sequence with known modifications, is there any methods to identify/discover and confirm the unknown modifications?

2.       In the manuscript, the authors detected m6A modification on two positions of HIV-1 RNA U3 region. Could this method also can detect the other modifications if it existed in the same RNase T1 digested oligonucleotide?

3.       Page 2, Lines 78 - 81, the description is not accurate? Because for any RNAs, tRNA, rRNA or even larger RNAs, like HIV-1 RNA, RNase T1 will digest RNA into oligonucleotide from 3’ of guanosine, therefore, if LC-MS is verified effective for tRNA and rRNA, it should also affective for HIV-1 RNA.

4.       Page 4, in method 2.4, the purified product includes DNA as well. Will the DNA affect oligonucleotide identification?

5.       Page 4, Lines 168 – 173, I don’t understand why the authors use different stages of detecting. Do they try exhaustive MS/MS to make sure more oligonucleotides are fragmented and identified?

6.       Please add limitation in discussion.

Author Response

Reviewer #1

Comments and Suggestions for Authors

In this manuscript, the authors developed a new method for targeted oligonucleotide modification identification to confirm the detection result of their previous DRS result on HIV-1 RNA. This is a brand new method for oligonucleotide modification identification, but the drawbacks are also obvious, this method is only suitable for known RNA sequence with known modification on known position. The followings are my comments:

Author response:

Oligonucleotide LC-MS/MS determines chemical modifications of RNAs with known sequences. The modification positions and the type of modifications, however, can be determined without prior information.

To clarify this issue, we modified our manuscript as follows:

New (Lines 75 – 77)

“This technology can determine chemical modifications in RNAs with known sequences. However, the positions and types of modifications can be identified without prior information[23-25].

  1. Except for the known RNA sequence with known modifications, is there any methods to identify/discover and confirm the unknown modifications?

Author response:

For an identification of an unknown modification (a new type of chemical modifications), one can employ a complete hydrolysis of RNA before LC-MS {Jora, 2021 #218}: Jora, M et al, (2021), Chemical Amination/Imination of Carbonothiolated Nucleosides During RNA Hydrolysis. Angew Chem Int Ed Engl 60, 3961-3966. This approach, however, does not provide modification site information.

For an identification modification sites, various RNA modification mapping tools have been published, including short-read sequencing tools. Unlike these tools, however, LC-MS/MS enables direct analysis of native RNA molecules.

To clarify this issue, we modified our manuscript as follows:

Introduction 1

Before (Lines 51 – 55)

Most studies to date have utilized short-read sequencing methods, involving reverse-transcription of fragmented RNA and PCR amplification of reverse-transcribed cDNA, providing only low-resolution and population-average values of chemical modifications. Short-read data are complicated by potential biases introduced during antibody-mediated target-RNA enrichment and cDNA preparation[6, 7].

New (Lines 51 – 56)

Most studies to date have utilized short-read sequencing methods, involving reverse-transcription of fragmented RNA and PCR amplification of reverse-transcribed cDNA, providing only low-resolution and population-average values of chemical modifications. While useful in identifying chemical modification sites without prior knowledge, short-read data are complicated by potential biases introduced during antibody-mediated target-RNA enrichment and cDNA preparation[6, 7].

Introduction 2

Before (Lines 61 – 64)

Mass spectrometry (MS) is a powerful tool capable of directly analyzing chemical modifications of native RNA molecules by measuring the mass shift of RNA oligomers[23, 24]. Oligonucleotide LC-MS/MS, in particular, identifies both the positions and types of chemical modifications at single-nucleotide resolution.

New (Lines 62 – 67)

Mass spectrometry (MS) is a powerful tool capable of directly analyzing chemical modifications of native RNA molecules by measuring the mass shift of RNA oligomers[23, 24]. To identify an unknown modification (a new type of chemical modification), one can employ complete hydrolysis of RNA before LC-MS[25]; however, this approach does not provide modification site information. Oligonucleotide LC-MS/MS, however, identifies both the positions and types of chemical modifications at single-nucleotide resolution.

  1. In the manuscript, the authors detected m6A modification on two positions of HIV-1 RNA U3 region. Could this method also can detect the other modifications if it existed in the same RNase T1 digested oligonucleotide?

Author response:

Potentially any RNA modifications can be detected with LC-MS/MS, if they are site-specific and deposited above the detection levels (to distinguish the mass shift between unmodified and modified RNAs). Nevertheless, we did not find evidence of additional modifications.

To clarify this issue, we modified our manuscript as follows:

New (Lines 320 – 322)

“While LC-MS/MS has the potential to detect any RNA modifications if they are site-specifically deposited above the detection levels, we did not detect any other modifications in our experiments.

  1. Page 2, Lines 78 - 81, the description is not accurate? Because for any RNAs, tRNA, rRNA or even larger RNAs, like HIV-1 RNA, RNase T1 will digest RNA into oligonucleotide from 3’ of guanosine, therefore, if LC-MS is verified effective for tRNA and rRNA, it should also affective for HIV-1 RNA.

Author response:

As you increase the length of RNA, the number of sequence repeats and sequence isomers proportionately increase. Because RNA is made from 4 nucleotides (unlike proteins made from 20 amino acids), the odds of getting similar nucleotide sequence and composition arising from multiple locations of RNA sequence also increase. Therefore, the true coverage obtained by unique nucleotide sequences (that match to only one location) decreases. For example, for 4.6 kb length covid spike protein mRNA, T1 can only provide <50% sequence coverage.

Nucleobase-specific RNases – including guanosine (G)-specific ribonuclease T1, cytidine (C)-specific Cusativin, or uridine (U)-specific MC1 ribonucleases – will cleave every G, C, or U, respectively, for the target RNA. Despite the frequent cleavage, smaller RNAs (tRNA, rRNA) can still be analyzed with Oligonucleotide LC-MS/MS because the RNase-cleaved fragments can be unambiguously identified based on mass differences among RNase-cleaved RNA fragments. Nevertheless, analyzing larger RNA molecules (e.g. 4.6 kb covid spike mRNA or 9 Kb HIV-1 RNA) is improbable. A nucleobase-specific RNase digestion of these long RNAs will generate RNA fragments with the same molecular weight, originating from different part of the long RNA. This prevents the unambiguous mapping of RNase-cleaved fragments.

To clarify this issue, we modified our manuscript as follows:

Before (Lines 78 – 81)

“While oligonucleotide LC–MS/MS has proven effective for small RNAs, such as transfer RNA (tRNA) and ribosomal RNA (rRNA)[7, 24, 31-37], analyzing larger RNAs, like the 9 Kb HIV-1 RNA, is nearly impossible due to the ambiguity in mass differences among RNase-cleaved fragments in LC-MS/MS data.”

New (Lines 84 - 91)

“Despite frequent cleavage with nucleobase-specific RNases, the analysis of small RNAs, such as transfer RNA (tRNA) and ribosomal RNA (rRNA) with oligonucleotide LC–MS/MS, has proven effective[7, 24, 31-37]. However, the analysis of larger RNAs, such as the 9 Kb HIV-1 RNA, becomes nearly impossible due to the ambiguity in mass differences among RNase-cleaved fragments in LC-MS/MS data. For example, RNase T1 digestion generates many RNA fragments with the same molecular weight, originating from different parts of the HIV-1 RNA. This prevents the unambiguous mapping of RNase T1-cleaved fragments.

  1. Page 4, in method 2.4, the purified product includes DNA as well. Will the DNA affect oligonucleotide identification?

Author response:

No. DNA oligos will not affect the results. Since DNA’s deoxyribose is smaller by 16 Da per nucleotide than RNA’s, DNA can be distinguished from RNA by mass. Moreover, the fragmentation pattern of oligonucleotide anions differs between DNA and RNA.

  1. Page 4, Lines 168 – 173, I don’t understand why the authors use different stages of detecting. Do they try exhaustive MS/MS to make sure more oligonucleotides are fragmented and identified?

First stage MS can only identify the mass of oligonucleotide not the nucleotide sequence. Second stage of MS helps identify the sequence based on sequence-informative fragment ion patterns.

To clarify this issue, we modified our manuscript as follows (yellow highlighted):

Before (Lines 168 – 173)

“A scan range of 545-2000 m/z (0.5 sec) was employed for the first stage (MS) data acquisition and 250-2000 m/z (1 sec) for the second (MS/MS) stage data acquisition. The top three most abundant ions in the first stage were chosen for fragmentation for MS/MS using an m/z dependent collision energy profile (20–23 V at m/z 545; 51–57 V at m/z 2000) before exclusion for 60 seconds using the dynamic exclusion feature.”

New (Lines 178 – 185)

“A scan range of 545-2000 m/z (0.5 sec) was employed for the first stage (MS) data acquisition and 250-2000 m/z (1 sec) for the second (MS/MS) stage data acquisition. The first stage of MS identifies the mass of the oligonucleotide, while the second stage helps identify the nucleotide sequence based on sequence-informative fragment ion patterns. The top three most abundant ions in the first stage were chosen for fragmentation for MS/MS using an m/z dependent collision energy profile (20–23 V at m/z 545; 51–57 V at m/z 2000) before exclusion for 60 seconds using the dynamic exclusion feature.”

  1. Please add limitation in discussion.

Author response:

The Discussion Section is modified as follows:

Before (Lines 332-339)

“           Our target-enrichment strategy notably improved the enrichment and purification of small target fragments of HIV-1 RNA containing both A8975 and A8989 sites. However, this process requires a substantial amount (~10 µg) of 9 Kb full-length virion RNAs to obtain an adequate amount of target RNA for LC-MS analysis. The current sensitivity achieved with this RNA (200 ng of target RNA) was adequate for detecting the presence of m6As at these two sites. Nevertheless, accurate quantification of m6A modifications would necessitate a significantly larger quantity of viral RNA and a well-designed quantification experiment.

New (Lines 349-369)

“           Our target-enrichment strategy notably improved the enrichment and purification of small target fragments of HIV-1 RNA containing both A8975 and A8989 sites. However, this process requires a substantial amount (~10 µg) of 9 Kb full-length virion RNAs to obtain an adequate amount of target RNA for LC-MS analysis. The current sensitivity achieved with this RNA (200 ng of target RNA) was adequate for detecting the presence of m6As at these two sites. Nevertheless, accurate quantification of m6A modifications would necessitate a significantly larger quantity of viral RNA and a well-designed quantification experiment. Method precision can be further improved by using another ribonuclease enzyme with complementary specificity so that overlapping digestion product is generated to confirm the modification location.

One limitation of this approach is its inability to distinguish nitrogenous base methylation from ribose methylation through oligonucleotide sequencing. This limitation may be addressed by incorporating nucleoside analysis, where base methylation and ribose methylation would exhibit different retention times during chromatography. The point of methylation can then be understood through MS/MS analysis of nucleosides.

While this method is effective in analyzing one target section, it has limitations, as it retains only the specified target, discarding the remaining sections during the process. Looking ahead, a multiplex technique capable of isolating multiple diverse targets from the same RNA sample would be valuable. Such an advancement could broaden the applicability of the approach, enabling the mapping of other regions of HIV-1 RNA or small segments within any long RNA molecules with known sequences.”

Reviewer 2 Report

Comments and Suggestions for Authors

The authors have performed excellent LC-MS/MS study on nucleotide resolution mapping of chemical modification sites on HIV-1 RNA. However, the current method proposed in the study requires larger sample quantity and needs further improvement in terms of sensitivity and precision. The manuscript is well written and demands further changes as under:

1.       Please list ways to improve the method sensitivity which is economical and precise for nucleotide mapping in the manuscript or include alternative detection methods to LC-MS/MS with comparable sensitivity and speed of detection if any, in the manuscript.

3.       Please include space between number and degree Celsius symbol throughout the manuscript.  Also, please keep the degree symbol in the manuscript consistent. Thank you.

Comments on the Quality of English Language

Minor formatting with regards to degree celsius symbol is required.

Author Response

Top of Form

Reviewer #2

Comments and Suggestions for Authors

The authors have performed excellent LC-MS/MS study on nucleotide resolution mapping of chemical modification sites on HIV-1 RNA. However, the current method proposed in the study requires larger sample quantity and needs further improvement in terms of sensitivity and precision. The manuscript is well written and demands further changes as under:

  1. Please list ways to improve the method sensitivity which is economical and precise for nucleotide mapping in the manuscript or include alternative detection methods to LC-MS/MS with comparable sensitivity and speed of detection if any, in the manuscript.

Author response:

Method precision can be obtained by using another ribonuclease enzyme with complementary specificity so that overlapping digestion product is generated to confirm the modification location.

Alternative methods to LC-MS/MS include modification-specific sample treatment and NGS-based methods following conversion of RNA to cDNA through reverse transcription. Modification presence is known by the RT skips or misincorporations (see Introduction lines 51-61).

Following statement is included in the Discussion Section

New (Lines 356-358)

Method precision can be further improved by using another ribonuclease enzyme with complementary specificity so that overlapping digestion product is generated to confirm the modification location.

  1. Please include space between number and degree Celsius symbol throughout the manuscript. Also, please keep the degree symbol in the manuscript consistent. Thank you.

Author response:

Thank you. The manuscript is modified accordingly.

Comments on the Quality of English Language

Minor formatting with regards to degree celsius symbol is required.

Author response:

Thank you. The manuscript is modified accordingly.

Reviewer 3 Report

Comments and Suggestions for Authors

Baek et collegues developed a protocol to map chemical modification on target fragment of RNA. They focused on the HIV-1 RNA to confirm the presence of two N6-methyladenosine predicted in 8975 and 8989 sites. They started isolating a quite big amount of virion RNA (about 10 ug) and applied an RNA fragments enrichment by the combination of S1 enzyme digestion and multiple non-target probes, follow by RNase T1 digestion. The resulted pool of fragments was suitable for LC-MS analysis and the comparison with synthetic RNA (positive controls) allowed the identification of the two modified sites.

The only limit of this approach is not being a shotgun approach, it is necessary to know the region where the chemical modification is expected. Nevertheless, is a great improvement in the study of oligonucleotide modification by mass spectrometry, a field that is still poorly studied.

Author Response

Reviewer #3

Comments and Suggestions for Authors

Baek et collegues developed a protocol to map chemical modification on target fragment of RNA. They focused on the HIV-1 RNA to confirm the presence of two N6-methyladenosine predicted in 8975 and 8989 sites. They started isolating a quite big amount of virion RNA (about 10 ug) and applied an RNA fragments enrichment by the combination of S1 enzyme digestion and multiple non-target probes, follow by RNase T1 digestion. The resulted pool of fragments was suitable for LC-MS analysis and the comparison with synthetic RNA (positive controls) allowed the identification of the two modified sites.

The only limit of this approach is not being a shotgun approach, it is necessary to know the region where the chemical modification is expected. Nevertheless, is a great improvement in the study of oligonucleotide modification by mass spectrometry, a field that is still poorly studied.

Author response:

Thank you. Following statement is included in the Discussion Section

New (Lines 364-369)

While this method is effective in analyzing one target section, it has limitations, as it retains only the specified target, discarding the remaining sections during the process. Looking ahead, a multiplex technique capable of isolating multiple diverse targets from the same RNA sample would be valuable. Such an advancement could broaden the applicability of the approach, enabling the mapping of other regions of HIV-1 RNA or small segments within any long RNA molecules with known sequences.”
